# Investigation on Adsorption and Decomposition Properties of CL-20/FOX-7 Molecules on MgH_2_(110) Surface by First-Principles

**DOI:** 10.3390/molecules25122726

**Published:** 2020-06-12

**Authors:** Zhang Yang, Zhao Fengqi, Xu Siyu, Yang Fusheng, Yao Ergang, Ren Xiaobing, Wu Zhen, Zhang Zaoxiao

**Affiliations:** 1School of Chemical Engineering and Technology, Xi’an Jiaotong University, Xi’an 710049, China; yangzhang@stu.xjtu.edu.cn (Z.Y.); wuz2015@mail.xjtu.edu.cn (W.Z.); zhangzx@mail.xjtu.edu.cn (Z.Z.); 2Laboratory of Science and Technology on Combustion and Explosion, Xi’an Modern Chemistry Research Institute, Xi’an 710065, China; zhaofqi@163.com (Z.F.); siyu-zusy99@163.com (X.S.); yaoerg@126.com (Y.E.); 3Shanxi Northern Xing’an Chemical Industry CO.LTD, Taiyuan 030008, China; 18234065493@163.com

**Keywords:** CL-20, FOX-7, MgH_2_, First-Principles, energetic molecule

## Abstract

Metal hydrides are regarded as promising hydrogen-supplying fuel for energetic materials while CL-20 (Hexanitrohexaazaisowurtzitane) and FOX-7 (1,1-Diamino-2,2-dinitroethylene) are typical principal components commonly used in energetic materials. Hence, it is interesting to explore the interactions between them for development of new energetic systems. In this paper, the adsorption and decomposition of CL-20 or FOX-7 molecules on the MgH_2_ (110) crystal surface were investigated by employing the First-Principles. In total, 18 adsorption configurations for CL-20/MgH_2_ (110) and 12 adsorption configurations for FOX-7/MgH_2_ (110) were considered. The geometric parameters for the configurations, adsorption energies, charge transfer, density of states, and decomposition mechanism were obtained and analyzed. In most of the configurations, chemical adsorption will occur. Moreover, the orientation of the nitro-group in CL-20 or FOX-7 with regard to the MgH_2_ (110) surface plays an important role on whether and how the energetic molecule decomposes. The adsorption and decomposition of CL-20 or FOX-7 on MgH_2_ could be attributed to the strong charge transfer between Mg atoms in the first layer of MgH_2_ (110) surface and oxygen as well as nitrogen atoms in the nitro-group of CL-20 or FOX-7 molecules.

## 1. Introduction

With the development of hydrogen energy, various hydrogen storage materials emerge such as metal hydrides. One notable application of metal hydrides is the additive in energetic materials such as solid propellants and explosives [1,2,3,4,5,6]. Metal hydrides, which are commonly considered as energy carriers for the hydrogen economy [7], have also attracted attention in solid propulsion due to their high chemical energy and remarkable activity [8]. The hydrogen offered by metal hydrides via dehydrogenation reactions is found effective in reducing the relative molecular mass of gas products of combustion, which favours the increase of specific impulses [9,10] for propellants. Furthermore, oxidation of the metal and H_2_, which are the products of the dehydrogenation reaction, could also provide a large amount of energy. The metal hydrides could be added into propellants as high energy combustion agents, and the energy level of propellant can be improved. In a previous experimental study by the authors [11], the combustion characteristics were found improved when adding ZrH_2_ in a double-base propellant. Similarly, the addition of hydrogen storage materials in explosives can increase the total energy of explosion [12], and significantly improve the explosive properties of emulsion explosives [13].

In the last few decades, considerable efforts have been devoted to the investigation of metal hydrides [9,14,15,16,17,18,19,20,21,22,23,24,25,26,27,28] as fuels in energetic materials. Among the various metal hydrides, MgH_2_ has the advantages of high hydrogen capacity (7.6wt.%.), abundant resources, little air pollutions after combustion, and good stability (dehydrogenation temperature of 300 ℃). Thus, it is paid great attention. The effect of MgH_2_ on thermal decomposition performance of cyclortrimethylenetrinitramine (RDX) was researched by Yao et al. [29]. Differential Scanning Calorimetry (DSC) was used to study the thermal decomposition characteristics of RDX with the addition of MgH_2_. The results show that MgH_2_ decreases the apparent activation energy of RDX from 159.22 kJ/mol to 133.69 kJ/mol. The thermal decomposition behavior of ammonium perchlorate (AP) in the presence of MgH_2_ was investigated through DSC by Liu et al. [30]. The results show that 5% MgH_2_ can decrease the low and high peak temperatures by 35 ℃ and 44.2 ℃ during thermal decomposition of AP, respectively, and increase the apparent heat release of AP from 0.44 kJ × g^−1^ to 1.20 kJ × g^−1^, which indicates a remarkable catalytic effect of MgH_2_ on AP thermal decomposition. The effect of MgH_2_ on thermal decomposition process of ammonium nitrate (AN) was studied by Wei et al. [31]. The results show that the addition of MgH_2_ makes the initial temperature of decomposition reduce greatly and the decomposition mechanism of AN change.

Although the introduction of metal hydrides into propellants and explosives shows great advantages, the interaction mechanism between metal hydrides and energetic compounds remains unclear due to the huge diversity of both components, which is one key issue restricting the application of metal hydrides in energetic materials. The energy performance of energetic materials is difficult to control without sufficient understanding about the interaction of the concerning components, which causes hidden risks when preparing, using, and storing energetic materials. Unfortunately, many current studies are focused on the effects of metal hydrides addition on the properties of energetic materials through experiments [11,12,13,29,30,31], which are costly and hazardous. In this sense, the numerical simulation based on theoretical model provides a reasonable tool to study the interactions among various components in energetic materials. For example, the adsorption and decomposition properties of explosive molecules on the metal surface have been studied theoretically by some scholars. The adsorption and decomposition of RDX, HMX (octahydro-1,3,5,7-tetranitro-1,3,5,7-tetrazocine), and CL-20 molecules on the Al (111) surface [32,33] or FOX-7 on Al_13_ clusters [34] were investigated by Ye et al. employing DFT (density functional theory) calculations. The adsorption and decomposition of the Nitroamine (NH_2_NO_2_) molecule on Al (111) and Mg (001) surface were studied by Zhou et al. [35,36] from the simulation results of adsorption energy, charge transfer, and adsorption energies of adsorption configurations.

In this paper, the interactions of two typical principal components in energetic materials, i.e., CL-20 and FOX-7, with magnesium hydride were studied by exploring the adsorption and decomposition properties of the energetic molecules on the MgH_2_(110) surface. The adsorption configurations, adsorption energies, charge transfer, and density of states before and after the adsorption were calculated by First-Principles simulation, in hope of shedding light on how and why the concerning energetic molecules interact with MgH_2_, as well as its significance in preparing for propellants with added metal hydrides.

## 2. Calculation Method and Model

### 2.1. Calculation Method

All calculations were performed by employing the CASTEP (Cambridge Sequential Total Energy Package) program [37] with Vanderbilt-type ultrasoft pseudo potentials [38] and a plane-wave expansion of the wave functions in the software package Materials Studio 8.0. General gradient approximation (GGA) was adopted in the exchange and correlation interactions. PBE (the functional form proposed by Perdew, Burke, and Ernzerhof [39]) was employed. The electronic wave functions were obtained by a density-mixing scheme [40] and the structures were relaxed using the Broyden, Fletcher, Goldfarb, and Shannon method [41]. The cut-off energy was set as 380 eV and the k-point sampling was set as 2 × 2 × 1, which showed good convergence for energy, geometry, and force. When the interatomic interaction force is less than 0.05 eV/Å, the stress is less than 0.1 GPa, the change in atomic energy is less than 2.0 × 10^−5^ eV/atom, and the change in displacement is less than 0.002 Å. The condition of convergence was deemed to be met. Spin polarization was not considered in the calculation.

### 2.2. Computational Model

MgH_2_ has three crystal morphologies [42,43], and the α-form among them is the most stable at normal temperature, whose (110) face is most stable [44]. Therefore, the computational model employed a 4 × 2 × 1 supercell and a nine-layer MgH_2_ (110) surface (as shown in Figure 1). The cell size with a rhombic box of a × b × c was 12.90 Å × 12.04 Å × 30.15 Å.

Five polymorphs, α to ε, are known for CL-20. The ε-polymorph is stable at room temperature and shows the highest density [45]. One ε-CL-20 molecule [46,47] (as in Figure 2) was placed on the upper side of the MgH_2_ (110) surface. Two types of nitro groups exist in the CL-20 molecule. The one attached to the six-member ring of CL-20 was represented by type-A, and the other attached to the five-member ring of CL-20, which was represented by type-B (see Figure 2).

1,1-Diamino-2,2-dinitroethylene (FOX-7) is a novel high energetic ingredient [48], whose structure [49,50] was shown in Figure 3. One FOX-7 molecule was placed on the upper side of the MgH_2_ (110) surface. Furthermore, 22 Å was taken as the thickness of a vacuum layer in both systems.

## 3. Calculation Results and Discussion

### 3.1. Geometries Parameters

The main adsorption sites on the MgH_2_ (110) surface are shown in Figure 4, which include Mg-top, H-top, Mg-Mg bridge, H-H bridge, Mg-H bridge, and hole.

A total of 18 possible adsorption configurations of CL-20 molecules on the MgH_2_ (110) face were considered in the calculation. (1) Type-A nitro group in CL-20 is adsorbed on the MgH_2_ (110) surface, and the configurations for the above six sites are signed (a)~(f). (2) Type-B nitro group in CL-20 is adsorbed with the nitro bond perpendicular to the MgH_2_(110) surface, and the configurations for the six sites are signed (g)~(l). (3) Type-B nitro group in CL-20 is adsorbed with the nitro bond parallel to the MgH_2_ (110) surface, and the configurations for the six sites are signed (m)~(r).

Similarly, 12 possible adsorption configurations for FOX-7 molecules to adsorb on the MgH_2_ (110) surface were considered. (1) The nitro group is adsorbed with a nitro bond perpendicular to the MgH_2_ (110) surface, and the configurations for the six sites are signed F-V1, F-V2, F-V3, F-V4, F-V5, and F-V6. (2) The nitro group is adsorbed with the nitro bond parallel to the MgH_2_ (110) surface, and the configurations for the six sites are signed F-P1, F-P2, F-P3, F-P4, F-P5, and F-P6.

Table 1 shows some important geometrical parameters of the 18 CL-20/MgH_2_ configurations, where *r*_(N1–O1)_ is the bond length between N1 and O1 atoms, and a similar notation applies to *r*_(N1–O2)_, *r*_(N1–N2)_, *r*_(N3–O3)__,_
*r*_(N3–O4)_, *r*_(N3–N4)_. Before the adsorption, *r*_(N1–O1)0_ = 1.246 **Å**, *r*_(N1–O2)0_ = 1.245 **Å**, *r*_(N1–N2)0_ = 1.416 **Å**, *r*_(N3–O3)0_ = 1.248 **Å**, *r*_(N3–O4)0_ = 1.253 **Å**, *r*_(N3–N4)0_ = 1.393 **Å**.

It can be seen that the closer the nitro group of CL-20 is to the MgH_2_ (110) surface, the easier it will be to get the corresponding bonds elongated or ruptured. Besides, there are no bonds rupture or formation in the b, c, d, f, and k configurations, and physical adsorption are deemed to occur. For the rest of the configurations with decomposition, the bond rupture occurs mostly in the mono-N-NO_2_.

The geometrical parameters of the 12 FOX-7/MgH_2_(110) configurations are shown in Table 2, where similar notations for the bond length in the CL-20/MgH_2_ system applies to *r*_(N1–O1)_, *r*_(N1–O2)_, and *r*_(C1__–N1)_ in the FOX-7/MgH_2_ system. Before the adsorption, *r*_(N1__–O1)0_=1.242 **Å**, *r*_(N1__–O2)0_=1.250 **Å**, and *r*_(C1__–N1)0_=1.416 **Å**. Rupture of the bonds suggesting chemical adsorption occurs in the F-V1, F-V2, F-V5, F-V6, F-P2, and F-P5 configurations. Instead, the physical adsorption occurs in the rest of the six configurations. The bond rupture mainly occurs in the mono-nitro-N-O bond. This is followed by the bis-nitro-N-O bond, and the C-N bond does not rupture at all, which illustrates an increasing stability of the corresponding bonds. In general, the bonds of the FOX-7 molecule show more tendency to rupture the FV-type configurations, which exhibits an impact of nitro-bond orientation on the interaction between FOX-7 and MgH_2_.

### 3.2. Adsorption Energies

The adsorption energies (*E*_ad_) of CL20 and FOX-7 molecules on the MgH_2_ (110) surface were calculated. *E*_ad_ is defined as:*E*_ad_ = *E*_slab/molecule_ − (*E*_slab_ + *E*_molecule_) 
where the *E*_slab/molecule_ is the total energy of the adsorption configurations after adsorption. *E*_slab_ is the single point energy of the MgH_2_ (110) surface and *E*_slab_ = −48267.4eV. The *E*_molecule_ is the single point energy of the energetic material molecule, *E*_CL-20_ = −9492.8eV, *E*_FOX-7_ = −3198.4eV.

For the CL-20/MgH_2_ (110) system, adsorption energies *E*_ad_ for the 18 adsorption configurations are shown in Figure 5.

In the case of (f) configuration (type-A nitro vertical to hole), whose adsorption energy is the lowest (−10.5 eV), no bond rupture or formation occurs and, hence, physical adsorption is expected. However, in the case of (n) configuration (type-B nitro parallel to H top), whose adsorption energy is the highest (−21.8 eV), the mono-N-NO_2_ bond and mono-nitro mono-N-O bond of the type B nitro group rupture, producing NO_2_, oxygen atom, and CL-20 fragment. The greater the adsorption energy is, the more intense the corresponding interaction will be. Moreover, we can see that, at the same adsorption sites, the adsorption energies of configurations with type-B nitro adsorbed are generally larger than the configurations with type-A nitro adsorbed. It means that the type-B nitro are easier to adsorb on the MgH_2_ (110) surface than type-A nitro. For type-B nitro of CL-20, at the same adsorption sites, the adsorption energies of adsorption configurations where adsorbed nitro of CL-20 is parallel to MgH_2_ (110) are larger than the adsorption configurations where adsorbed nitro is perpendicular to MgH_2_ (110). It means that CL-20 molecule is easier to adsorb on the surface of MgH_2_ (110) when its nitro group is placed horizontally than being placed vertically.

Meanwhile, the adsorption energies *E*_ad_ of FOX-7/MgH_2_ (110) adsorption configurations are shown in Figure 6. The negative adsorption energies for all the configurations, similar to the CL-20/MgH_2_ system, indicate exothermic and stable adsorption [51]. At the six adsorption sites, the adsorption energies of the FV-type configurations are unanimously greater than those of the FP-type configurations, which corresponds to a more stable adsorption. The highest adsorption energy is −21.2 eV when the nitro is vertical to the Mg-H bridge (F-V5 configuration), and the lowest adsorption energy is –15.9 eV when the nitro is parallel to the H-H bridge (F-P3 configuration).

### 3.3. Charge Transfer of Adsorption Configurations

Electron delocalization and charge transfer induce a chemical reaction of a system. In this section the charge transfer between Mg, H atoms in the first layer of the MgH_2_ (110) crystal face, and the activation centers of O and N atoms in the CL-20 or FOX-7 molecules is analyzed. Table 3 lists Mulliken charge distributions of the four types of atoms before and after adsorption. Before adsorption, Mg, H, O_A_ (the O atoms in type A nitro), O_B_ (the O atoms in type B nitro), N1, and N3 atoms have charge distribution ranges of 0.98 ~ 1.20e, −0.60 ~ −0.60e, −0.34 ~ −0.32e, −0.38 ~ −0.32e, 0.52e, and 0.52e, respectively. After adsorption, the charges of O_A_ and N1 atoms become significantly more negative in (a)~(f) configurations, while the charges of O_B_ and N3 atoms become significantly more negative in (g)~(r) configurations, and the charge of Mg atoms is significantly more positive, while the charge of H atoms shows little change, which indicates that strong charge transfer mainly occurs from Mg to O and N atoms in the adsorbed nitro group.

Figure 7 further shows the variation in the Mulliken charge (ΔCharge) of Mg and H atoms in the first layer of the MgH_2_(110) crystal face as well as O and N atoms in the CL-20 molecule before and after adsorption. The average charge of Mg atoms, H atoms, O atoms, and N atoms are, respectively, 1.0960e, −0.6000e, −0.3289e, and 0.5266e before adsorption (see the dashed lines in the figure). As can be found, in configurations (a)~(f), the average charge of Mg atoms increases by 0.0296 ~ 0.1440e, the average charge of H atoms decreases by 0.0100 ~ 0.0306e, the average charge of O_A_ atoms decreases by 0.2150 ~ 0.3075e, and the average charge of N1 atoms decreases by 0.4355 ~ 0.5715e. In configurations (g)~(r), the average charge of Mg atoms increases by 0.1284 ~ 0.1615e, the average charge of H atoms decreases by 0.0244 ~ 0.0350e, the average charge of O_B_ atoms decreases by 0.1001 ~ 0.2375e, and the average charge of N3 atoms decreases by 0.2782 ~ 0.55467e. Charge transfer between Mg in the first layer of the MgH_2_ (110) crystal face and O, N atoms of the adsorbed nitro group in CL-20 can be confirmed, which leads to the bond rupture in the CL-20 molecule.

The charge distribution of Mg and H atoms in the first layer, and the activation center O1, O2, and N1 atoms in the FOX-7 molecule of the FOX-7/MgH_2_ (110) system before and after adsorption are shown in Table 4. It can be seen that the charge of Mg atoms is increased after adsorption, while the charges of O1, O2, and N1 atoms are decreased. However, the charges of H atoms and C1 atoms show little changes. Apparently, strong charge transfer mainly occurs between Mg atoms and O, N atoms of the adsorbed nitro group in the FOX-7 molecule.

### 3.4. Density of States of Adsorption Configurations

In order to further investigate the interaction mechanism of the CL-20 or FOX-7 molecule with MgH_2_, the density of states (DOS) and partial density of states (PDOS) of the involving systems were analyzed. The DOS of MgH_2_, CL-20, and 18 configurations of CL-20/MgH_2_ (110), were shown in Figure 8. First, MgH_2_ shows DOS around two energy levels, which include −44.7eV ~ −40.3 eV and −9.2 eV ~ 4.2 eV. The results are close to those reported in Reference [52], verifying the slab model and parameters used. Second, the DOS for the 18 MgH_2_/CL-20 configurations are mainly distributed near three energy levels, −44.5 ~ −41.0 eV, −12.1 ~ −0.5 eV, and 0 ~ 5.5 eV. In proximity to the Fermi level, the region with the strongest DOS for CL-20 nearly completely coincides with the local DOS for MgH_2_, which indicates that the orbits of both are prone to mixing and hybridization. Therefore, the density of states near the Fermi level of all the 18 configurations has intensity significantly higher than those of CL-20 molecules or the MgH_2_ (110) crystal face.

Total density of states of Mg and H atoms in MgH_2_ (110) slab and the O and N atoms of CL-20 before adsorption are shown in Figure 9. The DOS after adsorption in configurations (g) are shown in Figure 10. We can see that the three peaks of Mg atoms near −44.5 ~ −40.5eV level are merged into one peak because the adsorbed CL-20 molecule causes the breakage of the original periodicity of the crystal structure of magnesium hydride. In addition, the DOS of O and N atoms move toward the lower energy level. Further analysis proves that the density of states of the Fermi level is mainly contributed by p orbits of Mg atoms, p orbits of N atoms, and O atoms of CL-20 molecules. The mixing and hybridization effect of the p orbits enhances the electron delocalization, promotes charge transfer, and, ultimately, leads to decomposition of CL-20 molecules on the MgH_2_ (110) crystal face.

In order to investigate why the decomposition of FOX-7 on the MgH_2_(110) surface is mainly caused by N-O bond rupture, its partial density of states (PDOS) was further investigated. The PDOS of Mg, H, O1, O2, and C1 atoms in F-V1 configurations are shown in Figure 11.

According to PDOS, it can be seen that the p orbital energy of Mg atoms and C1 atoms are located on both sides of the Fermi level, respectively, while the p orbital energy of O1, O2, and N1 atoms crosses the Fermi level. Apparently, Mg, O1, O2, and N1 atoms all have peaks in the vicinity of the Fermi energy levels. In addition, The DOS of Mg, O1, O2, and N1 orbital hybridization is likely to occur, which leads to strong interactions between Mg, O1, O2, and N1 atoms and promotes FOX-7 molecule adsorption on the surface of magnesium hydride. This also confirms the strong charge transfer between Mg and O, N mentioned previously.

### 3.5. Decomposition Mechanisms

Based on the previously mentioned calculations, the decomposition pathways of CL-20 or FOX-7 molecules on the MgH_2_(110) surface were obtained.

(Ⅰ) CL-20 decomposition mechanism

In total, five different decomposition mechanisms have been found for CL-20 (see Figure 12). Since there is more charge transfer between N atoms and Mg atoms than between O atoms and Mg atoms, N-NO_2_ is more likely to be ruptured than N-O of nitro when the CL-20 molecule is absorbed on the surface of MgH_2_(110).

(1) The mono-N-NO_2_ bond of type A nitro rupture involves the N-NO_2_ bond (attaching to type A nitro) in the symmetric position of N1-N2 bond being ruptured, which produces an NO_2_ and CL-20-1 fragment. This is applicable to the adsorption configurations of (a), and (e).

(2) The bis-N-NO_2_ bond of type B nitro rupture involves the bis-N-NO_2_ bond of type B nitro being ruptured, which produces two NO_2_ fragments and one CL-20-2 fragment. This is applicable to the adsorption configurations of (m), (o), and (p).

(3) The mono-N-NO_2_ bond of type B nitro and mono-nitro mono-N-O bond of type B nitro rupture involves mono-nitro mono-N-O bond of type B nitro being ruptured and producing O. The mono-N-NO_2_ bond of type B nitro is ruptured, which produces an NO_2_ and CL-20-3 fragment. This is applicable to the adsorption configurations of (n).

(4) The mono-N-NO_2_ bond of type B nitro rupture includes the mono-N-NO_2_ bond of type B nitro being ruptured, which produces the NO_2_ and CL-20-4 fragment. This is applicable to the adsorption configurations of (h), (j), (l), (q), and (r).

(5) The mono-nitro mono-N-O bond of type B nitro rupture includes the mono-nitro mono-N-O bond of type B nitro being ruptured, which produces the O and CL-20-5 fragment. This is applicable to the adsorption configurations of (g) and (i).

(Ⅱ) FOX-7 decomposition mechanism

For the decomposition of FOX-7 on the MgH_2_(110) surface, three different pathways were found. This is shown in Figure 13.

(1) The mono-nitro mono-N-O bond rupture involves the FOX-7-1 fragment and one oxygen atom being formed, which is applicable to F-V1, F-V2, F-V5, and F-P2 configurations.

(2) The mono-nitro bis-N-O bonds, mono-nitro mono-N-O bonds, and mono N-H bond rupture. The FOX-7-2 fragment, one oxygen atom, and one OH are formed, which is applicable to F-V6 configurations.

(3) Bis-nitro mono-N-O bonds rupture. The FOX-7-3 fragment and two oxygen atoms are formed, which is applicable to F-P5 configurations.

In addition, the T-Jump/FTIR combined technology was used to study the thermal decomposition of CL-20/MgH_2_ and FOX-7/MgH_2_ mixtures with a mass ratio of 1. The microscopic images of MgH_2_ samples were obtained (Figure 14) using the field emission scanning electron microscope Carl Zeiss SIGMA. MgH_2_ shows a regular spherical shape close to the morphology reported in Reference [29]. The analysis of gas-phase products was carried out using a fast scanning Fourier transform infrared spectrometer (Nicolet 5700FTIR). The interference pattern of incident light was obtained by the Michelson interferometer. The spectral data range was 650 to 4 000 cm^−1^. The time interval of rapid scanning thermal decomposition process data was 0.125 s. A high-purity argon atmosphere and normal pressure were applied in the experiments. The rapid thermal cracking process of the mixture of CL-20/MgH_2_ and FOX-7/MgH_2_ were studied under different temperatures and a heating rate of 10 K/min. The NO_2_ content in the product of CL-20/MgH_2_ and FOX-7/MgH_2_ with temperature is shown in Figure 15. At a low temperature under 510.1K, the mixture produces NO_2_ with steady low content (less than 5 × 10^−13^). Beyond 510.1K, the content of NO_2_ increases sharply. The peak value of CL-20/MgH_2_ mixture is 2.1 × 10^−12^ at 540.7 K, and FOX-7/MgH_2_ mixture is 1.9 × 10^−12^ at 594.7 K. CL-20/MgH_2_ mixture can produce more NO_2_ at a lower temperature than the FOX-7/MgH_2_ mixture, which shows higher reactivity for thermal decomposition.

Combining the simulation and experimental results, better stability of the FOX-7/MgH_2_ than CL-20/MgH_2_ can be confirmed, which is consistent with the stability comparison between FOX-7 and CL-20 [53,54], and may be attributed to the special π-packing structure and hydrogen bonds of FOX-7 [49]. Furthermore, as pointed out in several previous studies [55,56,57], CL-20 decomposition generally starts from the fracture of the weakest bonds, i.e., mono-N-NO_2_, particularly those connecting the five-member ring and the type-B nitro group [58]. On the other hand, the C-N bond connecting the nitro group in the FOX-7 is relatively stable and not ready for rupture under normal conditions [59]. Such observations could account for the varying tendency of decomposition for adsorptions of type A and B nitro in CL-20 on the MgH_2_ (110) surface, and the different products from decomposition of CL-20/MgH_2_ and FOX-7/MgH_2_. 

## 4. Conclusions

The adsorption and decomposition of CL-20 or FOX-7 molecules on the MgH_2_(110) surface were studied in this paper using the First Principles method. The above research showed that:

(1) The bonds of the adsorbed nitro group in energetic molecules are either ruptured or elongated after adsorption, which corresponds to chemical or physical adsorptions, respectively. Negative adsorption energies for all the concerning configurations indicate exothermic and stable adsorption of CL-20 and FOX-7 molecules. The nitro groups attached to the five-member ring of CL-20 (type B nitro) are easier to adsorb on the MgH_2_(110) surface than the nitro group attached to the six-member ring of CL-20 (type A nitro). For the type-B nitro, the adsorption is easier to proceed when the corresponding nitro bond is parallel rather than perpendicular to the MgH_2_ (110) surface. On the other hand, chemical adsorption with bond rupture is less likely to take place for FOX-7 than for CL-20, and the configurations with adsorbed nitro perpendicular to the MgH_2_ (110) surface show more tendency for decomposition.

(2) The adsorption and decomposition of energetic molecules (CL-20 or FOX-7) on the surface of MgH_2_(110) is closely related to the strong charge transfer between Mg atoms in the MgH_2_(110) surface and oxygen as well as the nitrogen atoms in the adsorbed nitro group of energetic molecules. Meanwhile, through the DOS of Mg, O, and N, we have found that orbital hybridization is likely to occur near the Fermi energy level, which promotes adsorption of energetic molecules on the surface of MgH_2_(110) and the fracture of bonds thereafter.

(3) In total, five decomposition mechanisms of CL-20 on the surface of MgH_2_ (110) were determined for the 18 adsorption configurations under discussion in which the rupture of the mono N-NO_2_ bond is mostly involved and, hence, the main products contain NO_2_, oxygen atoms, and energetic molecule fragments. While for FOX-7/MgH_2_ (110) adsorption, three decomposition mechanisms of FOX-7 were found for the 12 adsorption configurations with the main products being oxygen atoms, OH, and FOX-7 fragments.

## Figures and Tables

**Figure 1 molecules-25-02726-f001:**
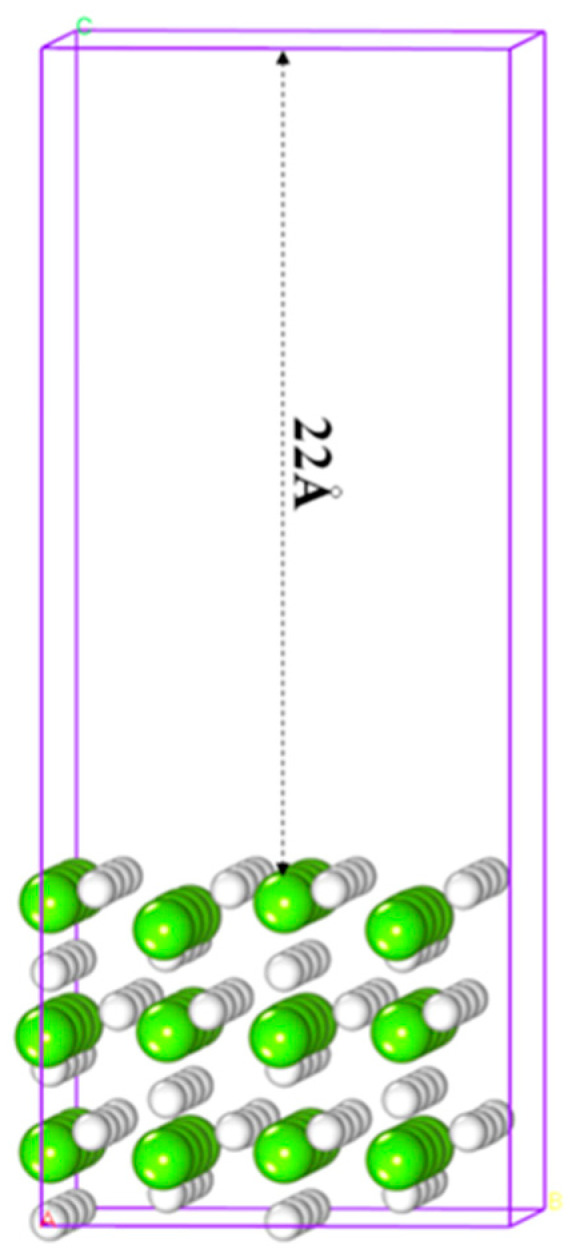
Slab model of MgH_2_ (110) where white and green are hydrogen and magnesium atoms, respectively.

**Figure 2 molecules-25-02726-f002:**
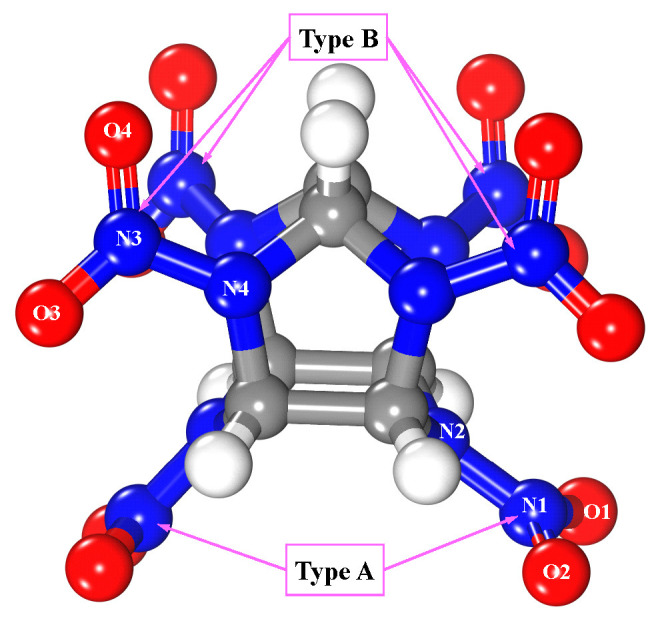
Structure of ε-CL-20 molecule, where white, blue, red, and gray spheres are hydrogen, nitrogen, oxygen, and carbon atoms, respectively.

**Figure 3 molecules-25-02726-f003:**
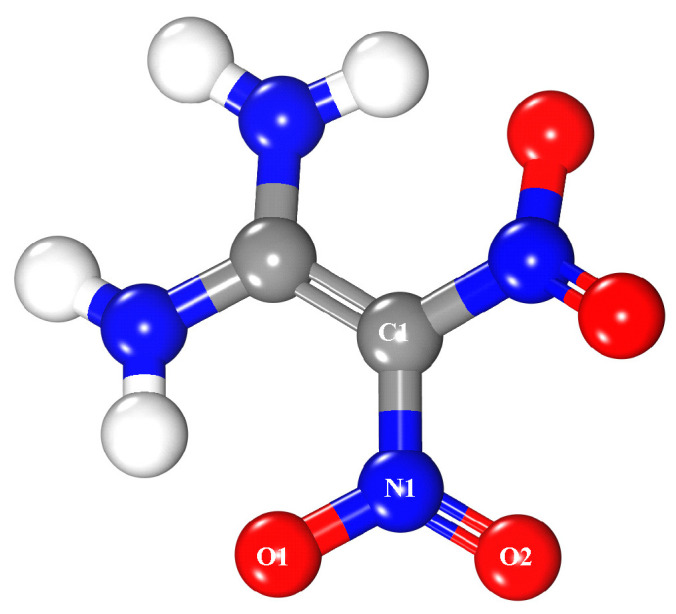
Structure of the FOX-7 molecule where white, blue, red, and gray spheres are hydrogen, nitrogen, oxygen, and carbon atoms, respectively.

**Figure 4 molecules-25-02726-f004:**
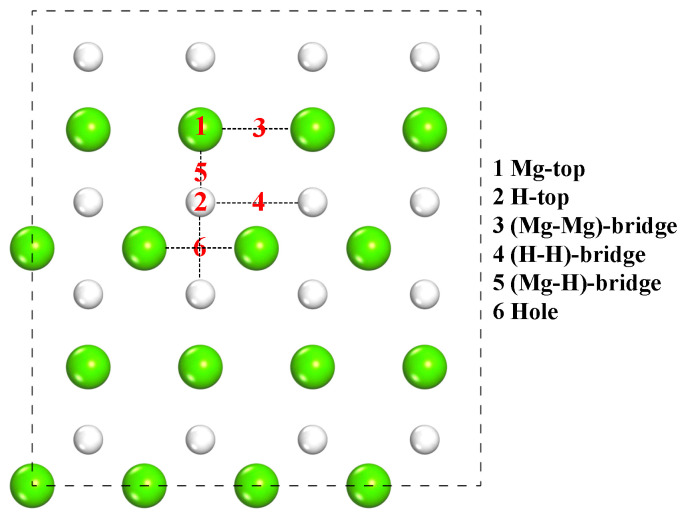
Schematic top view of initial adsorption sites of CL-20 or FOX-7 molecules on the MgH_2_ (110) surface.

**Figure 5 molecules-25-02726-f005:**
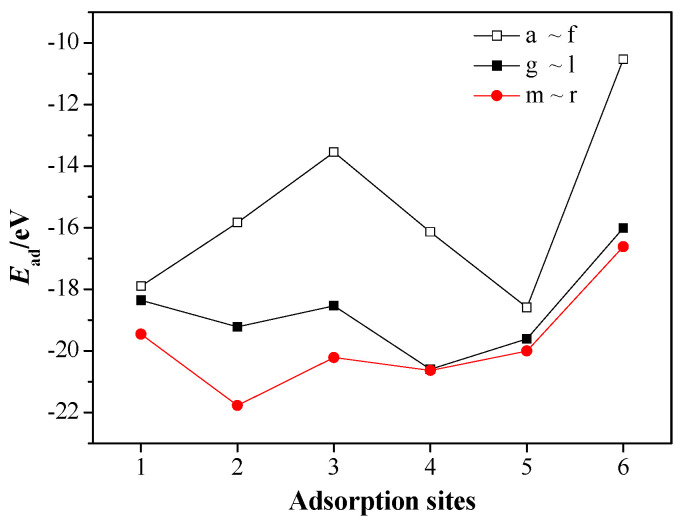
*E*_ad_ of the CL-20/MgH_2_ (110) configurations at different adsorption sites.

**Figure 6 molecules-25-02726-f006:**
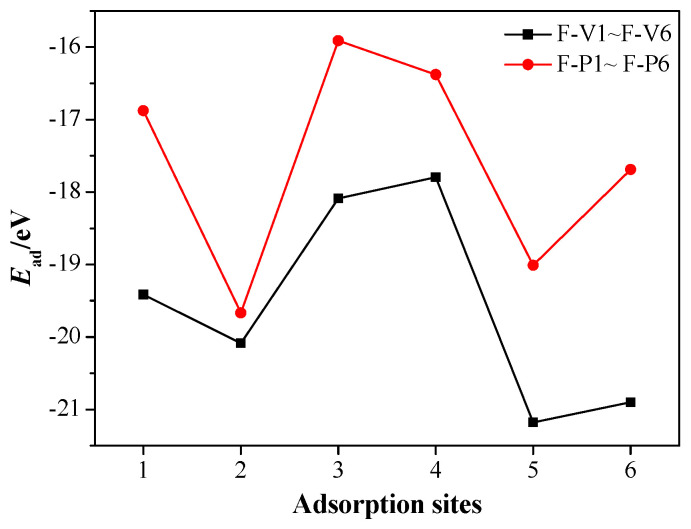
*E*_ad_ of the FOX-7/MgH_2_ (110) configurations at different adsorption sites.

**Figure 7 molecules-25-02726-f007:**
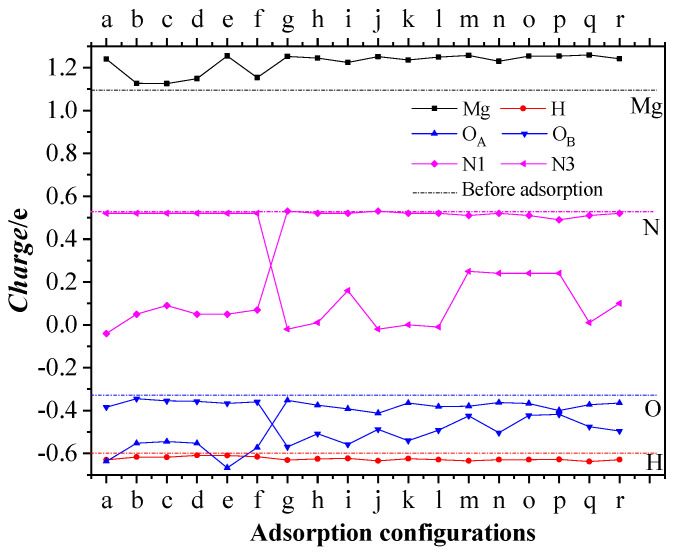
Variation in the average Mulliken charge of Mg and H atoms in the first layer of the MgH_2_(110) crystal surface as well as O (O_A_ and O_B_) and N (N1 and N3) atoms in the CL-20 molecule after adsorption. Dash lines are the charge before adsorption.

**Figure 8 molecules-25-02726-f008:**
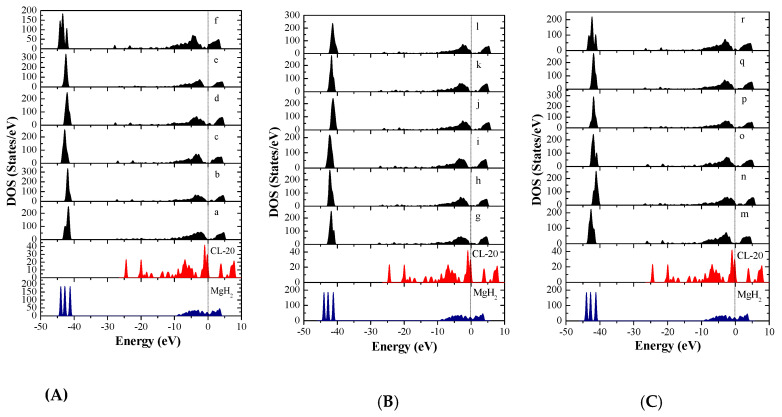
Total density of states (DOS) of the MgH_2_(110) crystal surface, CL-20 molecule, and CL-20/MgH_2_(110) configurations, with a Fermi level denoted by a vertical dashed line, (**A**) (a)~(f) configurations, (**B**) (m)~(r) configurations, and (**C**) (g)~(l) configurations.

**Figure 9 molecules-25-02726-f009:**
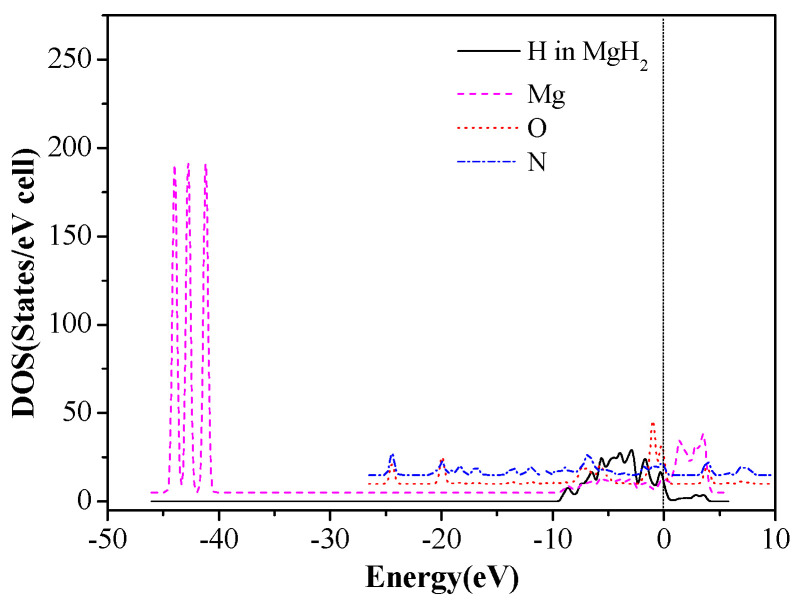
The DOS of Mg, H, O, and N before CL-20 adsorption on the MgH_2_ (110) surface.

**Figure 10 molecules-25-02726-f010:**
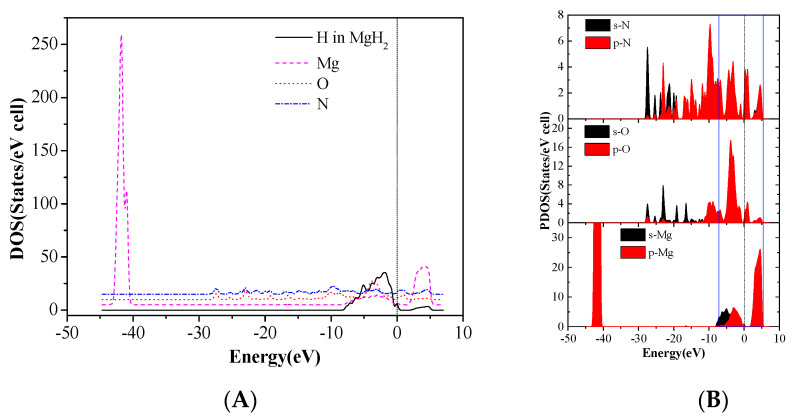
The DOS after adsorption in CL-20/MgH_2_(110) configurations (g), (**A**) sum, and (**B**) partial density of states (PDOS).

**Figure 11 molecules-25-02726-f011:**
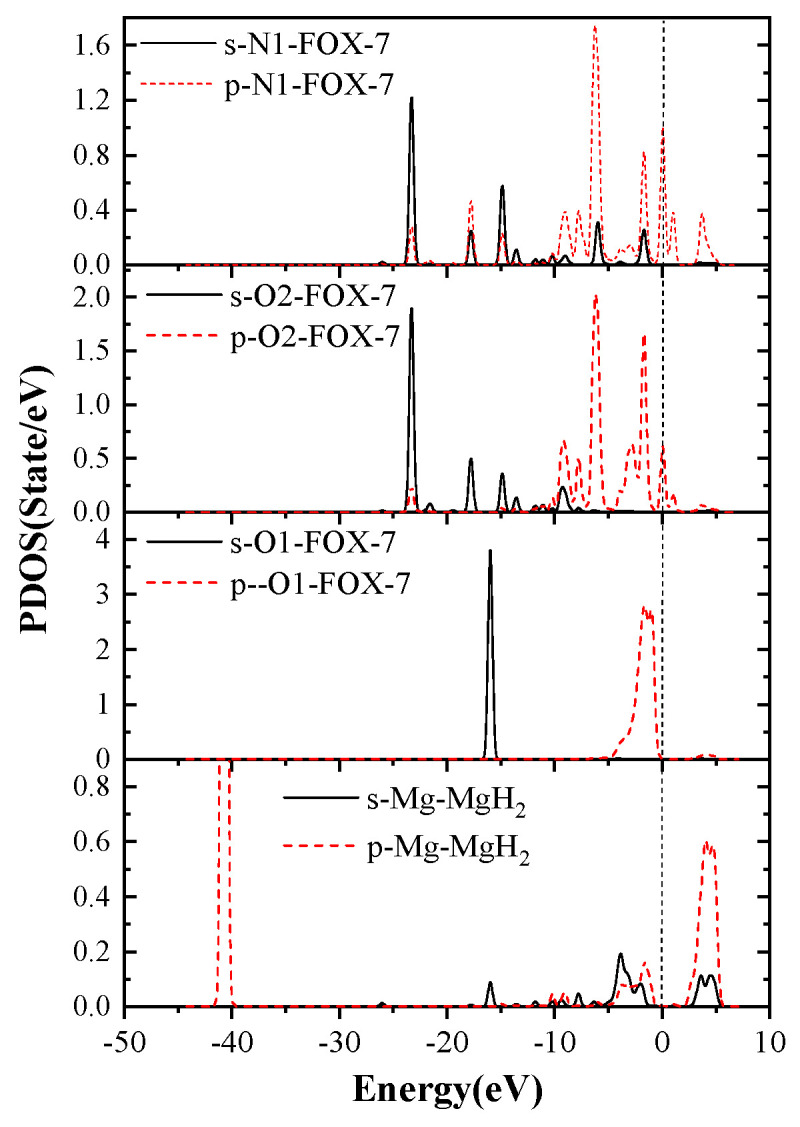
The PDOS of partial atoms in FOX-7/MgH_2_ (110) configurations (F-V1).

**Figure 12 molecules-25-02726-f012:**
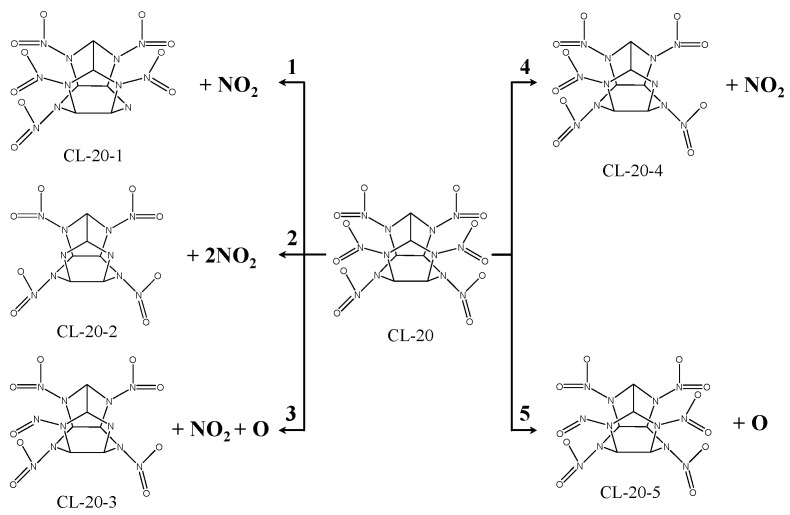
Graph of the decomposition mechanism of CL-20 molecules on the MgH_2_(110) face.

**Figure 13 molecules-25-02726-f013:**
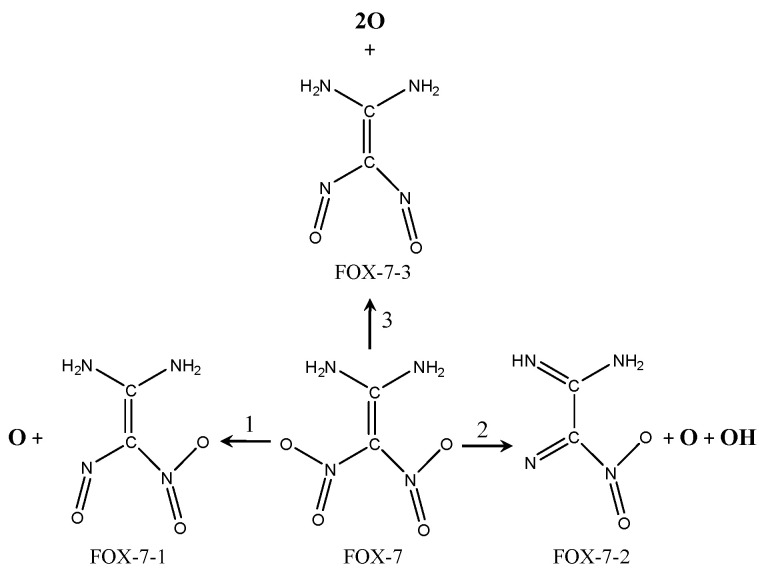
Graph of decomposition mechanism of FOX-7 molecules on the MgH_2_(110) surface.

**Figure 14 molecules-25-02726-f014:**
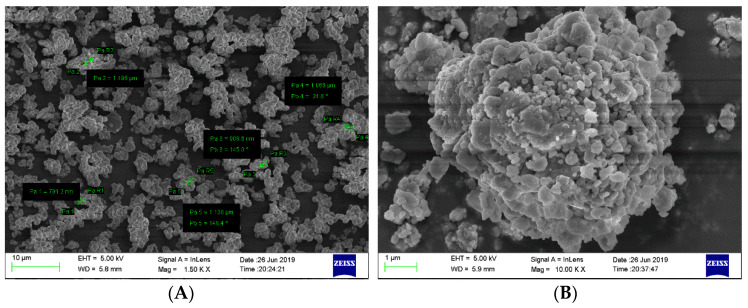
SEM images of MgH_2_ samples, (**A**) 1.50KX, and (**B**) 10.00KX.

**Figure 15 molecules-25-02726-f015:**
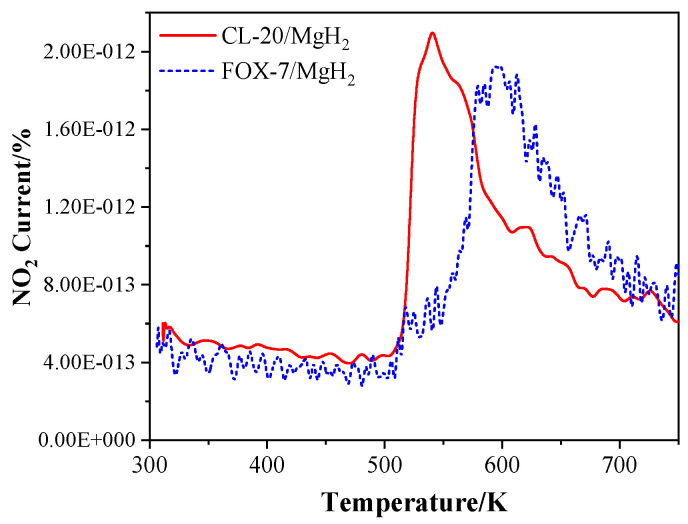
The curve of NO_2_ content in the product of the T-Jump/FTIR – MS (mass spectrometry) Analysis of CL-20 or FOX-7 and MgH_2_ with temperature.

**Table 1 molecules-25-02726-t001:** Geometrical parameters of the CL-20/MgH_2_(110) configurations after adsorption.

Adsorption Configuration	*r*_(N1-O1)_ (Å)	*r*_(N1-O2)_ (Å)	*r*_(N1-N2)_ (Å)	*r*_(N3-O3)_ (Å)	*r*_(N3-O4)_ (Å)	*r*_(N3-N4)_ (Å)
a	1.292	1.299	0	1.268	1.250	1.392
b	1.416	1.465	1.501	1.250	1.248	1.410
c	1.419	1.404	1.535	1.252	1.252	1.407
d	1.405	1.424	1.515	1.257	1.247	1.415
e	1.312	1.320	0	1.258	1.250	1.403
f	1.701	1.342	1.488	1.255	1.247	1.419
g	1.247	1.258	1.410	0	1.286	1.352
h	1.251	1.365	1.397	1.411	1.263	0
i	1.270	1.252	1.405	0	1.279	1.369
j	1.267	1.267	1.386	1.320	1.272	0
k	1.268	1.243	1.395	1.421	1.479	1.453
l	1.273	1.250	1.397	1.381	1.258	0
m	1.255	1.259	1.398	1.301	1.284	0
n	1.244	1.261	1.400	1.329	1.267	0
o	1.253	1.255	1.402	1.325	1.272	0
p	1.263	1.267	1.384	1.325	1.272	0
q	1.266	1.248	1.400	1.325	1.267	0
r	1.252	1.250	1.399	1.384	1.267	0

**Table 2 molecules-25-02726-t002:** Geometrical parameters of the FOX-7/MgH_2_ (110) configurations after adsorption.

Adsorption Configuration	*r*_(N1-O1)_ (Å)	*r*_(N1-O2)_ (Å)	*r*_(C1-N1)_ (Å)
F-V1	0	1.321	1.351
F-V2	0	1.505	1.385
F-V3	1.427	1.417	1.423
F-V4	1.450	1.458	1.429
F-V5	0	1.470	1.369
F-V6	0	0	1.233
F-P1	1.405	1.331	1.381
F-P2	0	1.340	1.330
F-P3	1.516	1.431	1.500
F-P4	1.496	1.435	1.475
F-P5	0	1.258	1.410
F-P6	1.477	1.450	1.483

**Table 3 molecules-25-02726-t003:** Charge distribution ranges of Mg and H atoms in the first layer, and the O_A_ and O_B_ atoms in the CL-20 molecule of the MgH_2_ (110)/CL-20 system before and after adsorption.

Adsorption Configuration	Mg Atoms in the First Layer(e)	H Atoms in the First Layer(e)	O_A_ Atom(e)	O_B_ Atom(e)	N1 Atom(e)	N3 Atom(e)
Before adsorption	0.98~1.20	−0.60~−0.60	−0.34~−0.32	−0.38~−0.32	0.52	0.52
a	0.96~1.58	−0.69~−0.57	−0.78~−0.49	−0.53~−0.34	0.04	0.52
b	0.59~1.40	−0.67~−0.58	−0.76~−0.34	−0.37~−0.32	0.05	0.52
c	0.56~1.41	−0.66~−0.59	−0.74~−0.36	−0.38~−0.33	0.09	0.52
d	0.63~1.48	−0.66~−0.54	−0.77~−0.34	−0.52~−0.32	0.05	0.52
e	1.01~1.77	−0.68~−0.57	−0.80~−0.56	−0.42~−0.34	0.05	0.52
f	1.04~1.42	−0.67~−0.59	−0.93~−0.36	−0.44~−0.32	0.07	0.52
g	1.04~1.46	−0.68~−0.57	−0.43~−0.32	−1.27~−0.32	0.52	0.00
h	1.05~1.36	−0.72~−0.58	−0.48~−0.33	−0.80~−0.34	0.52	0.01
i	0.96~1.41	−0.69~−0.58	−0.53~−0.33	−1.29~−0.33	0.52	0.16
j	0.99~1.57	−0.68~−0.59	−0.49~−0.34	−0.78~−0.34	0.52	0.01
k	1.05~1.48	−0.68~−0.56	−0.51~−0.29	−0.76~−0.33	0.52	0
l	1.03~1.77	−0.67~−0.58	−0.51~−0.33	−0.76~−0.34	0.52	0.01
m	1.01~1.50	−0.69~−0.55	−0.43~−0.34	−0.61~−0.34	0.51	0.25
n	1.04~1.45	−0.69~−0.55	−0.46~−0.31	−1.23~−0.32	0.52	0.24
o	0.97~1.56	−0.69~−0.58	−0.40~−0.34	−0.59~−0.33	0.51	0.24
p	1.01~1.47	−0.68~−0.59	−0.50~−0.30	−0.59~−0.33	0.49	0.24
q	1.08~1.48	−0.69~−0.59	−0.49~−0.31	−0.73~−0.33	0.51	0.01
r	0.98~1.58	−0.69~−0.59	−0.40~−0.34	−0.87~−0.33	0.52	0.10

**Table 4 molecules-25-02726-t004:** Variation in the average Mulliken charge of Mg and H atoms in the first layer of the MgH_2_ (110) crystal surface as well as O1, O2, and N1 atoms in the FOX-7 molecule before and after adsorption.

Adsorption Configuration	Mg Atoms in the First Layer(e)	H Atoms in the First Layer(e)	O1 Atom(e)	O2 Atom(e)	N1 Atom(e)	C1 Atom(e)
Before adsorption	1.10	−0.60	−0.42	−0.32	0.37	0.15
F-V1	1.21	−0.62	−1.27	−0.57	−0.15	0.17
F-V2	1.21	−0.61	−1.31	−0.87	−0.56	0.16
F-V3	1.15	−0.63	−0.71	−0.78	−0.03	0.17
F-V4	1.13	−0.62	−0.80	−0.79	−0.09	0.15
F-V5	1.23	−0.62	−1.26	−0.74	−0.53	0.18
F-V6	1.23	−0.58	−1.26	−1.13	−0.73	0.18
F-P1	1.11	−0.62	−0.70	−0.66	−0.13	0.19
F-P2	1.27	−0.65	−1.22	−0.62	−0.10	0.15
F-P3	1.24	−0.61	−0.79	−0.75	−0.06	0.16
F-P4	1.22	−0.63	−0.80	−0.76	−0.03	0.17
F-P5	1.21	−0.62	−1.29	−1.30	−0.06	0.19
F-P6	1.23	−0.73	−0.66	−0.87	−0.03	0.18

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
