# Peer review of "Investigation on Adsorption and Decomposition Properties of CL-20/FOX-7 Molecules on MgH2(110) Surface by First-Principles"

_molecules, 2020, doi:10.3390/molecules25122726_

Round 1

Reviewer 1 Report

The work deals with the first-principles investigation of energetic molecules adsorption on magnesium hydride surface and decomposition properties thereof.

I have to admit that this is not specifically my focus of research, I study metal hydrides for hydrogen storage purposes, but I think my opinion is valuable as scientist and as potential reader of the manuscript in this journal.

My overall opinion on the paper is quite negative, but this is not due to the merely scientific aspects (I cannot judge the soundness of the calculations and the results, because I am not familiar with the programs and packages used to all the calculations) but on how the whole work is organized, presented and written.

The abstract is too detailed and does not confer the main message of the work in a way stimulating any interest to read the work. The introduction is ok, it just need grammar/style/formatting revisions.

The model presentations could be simplified, adding some bullet point lists or table, in my opinion.

I do understand that the cases analyzed are numerous and they have to be reported, but then many paragraphs (especially in sections 3-4) are just lists of elements and the correlation between causes and effects is poorly underlined. A lot of text is used to explain the obvious meaning of labels.

The parts describing the content of the tables and then commenting the results listed there are hard to go through and do not help the reader grasping the main ideas and concepts. This is partially happening also in the figures descriptions and, most importantly, in the conclusions.

There are experiments performed with FTIR-MS mentioned in the text, but not a single experimental detail is provided, it is unclear how the tests were done and it would be impossible to repeat them.

Apart from the need for an extensive English grammar and style check, I see many formatting errors (subscripts, spacing). Generally, this does not reflect well on the quality of the work.

In conclusion, I think major revision are required before the work can be accepted for publication. As said in the beginning I am not saying that the investigation is not valuable, but some parts of the manuscript should be completely rewritten.

Author Response

Responses to reviewer #1

Questions or comments

Responses

The abstract is too detailed and does not confer the main message of the work in a way stimulating any interest to read the work.

We are sorry for the poor expression in the original manuscript. The abstract has been thoroughly rewritten and greatly shortened (about 50% in length), so that the main message of the manuscript can be clearly delivered to the readers.

The introduction is ok, it just need grammar/style/formatting revisions.

The introduction section was also revised to improve the English and format. Some less relevant sentences and paragraphs were removed or rewritten, so that the background and significance of the work can be better understood.

The model presentations could be simplified, adding some bullet point lists or table, in my opinion.

The model presentations were reorganized from the bullet points on MgH2, CL-20 and FOX-7 respectively in line 117-128, page 4.

I do understand that the cases analyzed are numerous and they have to be reported, but then many paragraphs (especially in sections 3-4) are just lists of elements and the correlation between causes and effects is poorly underlined. A lot of text is used to explain the obvious meaning of labels.

We appreciate the suggestions from the reviewer. We tried our best to remove or shorten the plain results or their obvious meaning in the main text (e.g. in sections 3 and 4), and further discussions were inserted to correlate the causes and effects, for example, in the last paragraph of section 3.5 in the revised manuscript.

The parts describing the content of the tables and then commenting the results listed there are hard to go through and do not help the reader grasping the main ideas and concepts. This is partially happening also in the figures descriptions and, most importantly, in the conclusions.

The parts mentioned by the reviewer were carefully checked, and the contents commenting the table, figure and in the conclusions, were reorganized to more clearly deliver the main ideas and concepts.

There are experiments performed with FTIR-MS mentioned in the text, but not a single experimental detail is provided, it is unclear how the tests were done and it would be impossible to repeat them.

We are sorry for the missing information. The description of experimental settings has been added in the line 321-330, page 12 of the revised manuscript.

Apart from the need for an extensive English grammar and style check, I see many formatting errors (subscripts, spacing). Generally, this does not reflect well on the quality of the work.

The English grammar and style have been double-checked with the help from Mrs. Fang Wang, who is a professor in the Department of English Language, Xi’an Jiaotong University. The authors also carefully checked the manuscript to remove the formating errors.

In conclusion, I think major revision are required before the work can be accepted for publication. As said in the beginning I am not saying that the investigation is not valuable, but some parts of the manuscript should be completely rewritten.

As shown in the marked version of the revised manuscript, it has been rewritten completely.

Reviewer 2 Report

This manuscript is an example for a theoretical work with the title „ Investigation of CL-20/FOX-7 molecule adsorption and decomposition properties on MgH2(110) surface by First-Principles“. The authors describe metal hydrides as potential hydrogen-supplying fuel for

energetic materials. With that two energetic materials are introduced for adsorption, CL-20 (Hexanitrohexaazaisowurtzitane) and FOX-7 (1,1-Diamino-2,2-dinitroethylene) and the surface adsorption and decomposition of CL-20/FOX-7 molecules on the

MgH2(110) crystal surface were investigated by employing the First-Principles.

The article is well written, but still has some minor English mistakes, which should be taken care of. It is worth to be published after major revisions. Please respond by a point by point answer to my points I mention below:

First of all, honestly, I read the whole thing and as an experimentalist, I am wondering why this is only a computational work? I will not ask for experiments during these times but I would advise you to have a bit more “flesh” for your next publication. In general, I feel that the introduction could use a proper literature update. Furthermore, a lot of spaces/ tabs are missing. The manuscript should be thoroughly revised.

Page 2: this sentence seems to cite various unrelated papers, especially for borohydrides you cite a work with closoborates?!. “In the last few decades, considerable efforts have been devoted to the investigation of metal hydrides, including aluminium hydride (AlH3)[9,14–16], berylliumhydride (BeH2)[17],magnesium hydride (MgH2) [18], titaniumhydride (TiH2) [19,20], lithium hydride (LiH)[21],ammonia borane(NH3BH3) [22], borohydrides[23], metal borohydride ammoniate[24] and zirconium hydride (ZrH2)[20],”

Especially the borohydrides should be dealt with more cautiously and the following should be added here: [1-3]

[1] Paskevicius M, Jepsen LH, Schouwink P, ÄŒerný R, Ravnsbæk DB, Filinchuk Y, et al. Metal borohydrides and derivatives - synthesis, structure and properties. Chem Soc Rev. 2017;46:1565-634.

[2] Frommen C, Sørby M, Heere M, Humphries T, Olsen J, Hauback B. Rare Earth Borohydrides—Crystal Structures and Thermal Properties. Energies. 2017;10:2115.

[3] Hadjixenophontos E, Dematteis EM, Berti N, WoÅ‚czyk AR, Huen P, Brighi M, et al. A Review of the MSCA ITN ECOSTORE—Novel Complex Metal Hydrides for Efficient and Compact Storage of Renewable Energy as Hydrogen and Electricity. Inorganics. 2020;8:17.

Page 6, line204: Å missing after the number r

Page 6, line206: a is larger than g, so you cannot say it like this. In general, why are there zeros in this table? Is that the chemisorption or chemical adsorption, meaning splitting of your molecule?

Page 67, line224: why is it r(n1-o1) here and r(n1-o1)1 in the table 2?

Page 10, line 309: Which lead the bond to repture in CL-20? Order of words makes it misleading.

Page 10, figure 7: there are horizontal and vertical dashed lines. Both are showing the charge of before adsorption?

Page 11, line 333: why you speak of the DOS for MgH2? before you mention two discrete DOS for MgH2.

Page 12: figure 10b, PDOS is mssing on Y-axis.

Page 12: figure 11, P-Mg-MgH2 – shoud that be a capital letter or not?

Page 13, line365: partial density of states should be introduced earlier.

Page 13, Figure 12 seems very unsharp or blurry – it seems copied from somehwere – if so, please add a reference (adapted from ... ) and the same applies to Figure 13.

Page 15: why is the FTIR not introduced in the experimental section?

Page 15, line 450-452: this sentence is a repetion from 435-437. Either delete or explain why its necessary to mention this twice.

There is a motivational sentence missing in the end why this work is important and if the research on these kind of materials will go on.

Author Response

Responses to reviewer #2

Questions or comments

Responses

First of all, honestly, I read the whole thing and as an experimentalist, I am wondering why this is only a computational work? I will not ask for experiments during these times but I would advise you to have a bit more “flesh” for your next publication.

The reviewer gives us very good suggestions. We would consider introducing more experimental contents to make the rigorous discussion possible in our future work.

In general, I feel that the introduction could use a proper literature update. Furthermore, a lot of spaces/ tabs are missing. The manuscript should be thoroughly revised.

The citation of the references were updated, and the spaces/ tabs were inserted as necessary in the introduction section. The manuscript was thoroughly revised as shown in the marked version.

Page 2: this sentence seems to cite various unrelated papers, especially for borohydrides you cite a work with closoborates?!. “In the last few decades, considerable efforts have been devoted to the investigation of metal hydrides, including aluminium hydride (AlH3)[9,14–16], berylliumhydride (BeH2)[17],magnesium hydride (MgH2) [18], titaniumhydride (TiH2) [19,20], lithium hydride (LiH)[21],ammonia borane(NH3BH3) [22], borohydrides[23], metal borohydride ammoniate[24] and zirconium hydride (ZrH2)[20],”

We appreciate the suggestions by the reviewer. Actually, we realize that the detailed reference to multiple kinds of metal hydrides one by one is unnecessary, and in the revised manuscript just a general reference to the publications [9, 14-28] is presented while focusing on the metal hydrides under discussion, i.e. MgH2.

Especially the borohydrides should be dealt with more cautiously and the following should be added here: [1-3]

[1] Paskevicius M, Jepsen LH, Schouwink P, ÄŒerný R, Ravnsbæk DB, Filinchuk Y, et al. Metal borohydrides and derivatives - synthesis, structure and properties. Chem Soc Rev. 2017;46:1565-634.

[2] Frommen C, Sørby M, Heere M, Humphries T, Olsen J, Hauback B. Rare Earth Borohydrides—Crystal Structures and Thermal Properties. Energies. 2017;10:2115.

[3] Hadjixenophontos E, Dematteis EM, Berti N, WoÅ‚czyk AR, Huen P, Brighi M, et al. A Review of the MSCA ITN ECOSTORE—Novel Complex Metal Hydrides for Efficient and Compact Storage of Renewable Energy as Hydrogen and Electricity. Inorganics. 2020;8:17.

The suggested references have been added in the revised manuscript as [24]-[26].

Page 6, line204: Å missing after the number r

Å were added where they are supposed to appear, in line 151-152, page 5 and line 161-162, page 6 of the revised manuscript.

Page 6, line206: a is larger than g, so you cannot say it like this. In general, why are there zeros in this table? Is that the chemisorption or chemical adsorption, meaning splitting of your molecule?

The paragraph right after Table 1 was reorganized and the sentence mentioned by the reviewer has been deleted in the revised manuscript. In our study, the chemisorption in accompanied with rupture of old bonds and formation of new ones, hence the zero values of bond length means the rupture of the bond and chemisorption.

Page 67, line224: why is it r(n1-o1) here and r(n1-o1)1 in the table 2?

The “r(N1-O1)1” notations in the Table 2 have been changed into “r(N1-O1)” to keep consistence with the text.

Page 10, line 309: Which lead the bond to rupture in CL-20? Order of words makes it misleading.

The strong charge transfer is thought to be closely related to the bond rupture in CL-20, the statement has been reorganized in line 228-230, page 8 of the revised manuscript.

Page 10, figure 7: there are horizontal and vertical dashed lines. Both are showing the charge of before adsorption?

The horizontal dashed lines show the charges of different atoms before the adsorption, a sentence has been inserted in line 222, page 8 to explain this. The vertical dashed lines were meant to show the division of Type-A and Type B nitro adsorption, which seems unnecessary and has been deleted to avoid misunderstanding.

Page 11, line 333: why you speak of the DOS for MgH2? before you mention two discrete DOS for MgH2.

Actually, two discrete DOS exist for MgH2 as mentioned in line 249-250, page 9 of the revised manuscript. However, please note here we refer to the DOS of MgH2 in the proximity to the Fermi level, and there is only one DOS there. We added “local” in line 254 of the revised manuscript to avoid misunderstanding.

Page 12: figure 10b, PDOS is missing on Y-axis.

We are sorry for the carelessness. The Y-axis of “PDOS” was added in Fig. 10(B).

Page 12: figure 11, P-Mg-MgH2 – shoud that be a capital letter or not?

The “P” was changed to “p” (lower case letter) in Fig.11 of the revised manuscript.

Page 13, line365: partial density of states should be introduced earlier.

The “DOS” and “PDOS” were introduced at the beginning of 3.4 (line 246-248, page 9) in the revised manuscript.

Page 13, Figure 12 seems very unsharp or blurry – it seems copied from somehwere – if so, please add a reference (adapted from ... ) and the same applies to Figure 13.

Figure 12 and 13 are actually prepared by ourselves, not copied from somewhere else. We have remade the figures as “.wmf” format and inserted them in the revised manuscript.

Page 15: why is the FTIR not introduced in the experimental section?

Actually, the experimental part of FTIR works locally for the comparison of decomposition easiess and products of CL-20/MgH2 and FOX-7/MgH2, in support of the First Principles study. Therefore, we did not introduce an individual section for the experimental details. However, in the revised manuscript, the description of experimental settings has been added in the line 321-330, page 12.

Page 15, line 450-452: this sentence is a repetion from 435-437. Either delete or explain why its necessary to mention this twice.

In response to the reviewer’s comment, the whole manuscript has been completely rewritten and we have tried our best to avoid any unnecessary repetion in the revised version.

There is a motivational sentence missing in the end why this work is important and if the research on these kind of materials will go on.

We thank the reviewers for the beneficial suggestions. Such motivational sentences describing the significance and necessity of the work was reorganized and could be found in line 73-77, 85-91 of the revised manuscript.

Reviewer 3 Report

There are many papers about decomposition process of HNIW and DADNE and the Authors should refer to them during discussion of their results.

Author Response

Responses to reviewer #3

Questions or comments

Responses

There are many papers about decomposition process of HNIW and DADNE and the Authors should refer to them during discussion of their results.

We are grateful to the suggestions by the reviewer. Some new references [54-59] on the decomposition of HNIW and DADNE were added in the revised manuscript, and the discussion about our simulation results was expanded in the last paragraph of section 3.5 referring to these previous work.

Round 2

Reviewer 1 Report

In my opinion, the overall structure and content of the paper benefited from the suggestions of all the reviewers and it is now improved and more easily readable/understandable. The authors added the missing experimental details and revised some parts according to the point that were listed in the previous report.

It was not very comfortable to go through the revised manuscript, with all the substituted/erased lines (I guess the editorial office should have provided a "clean" version as well), but that made clear that major revision were done.

In my opinion the paper can be now considered for publication, according to the other reviewers' suggestion and the decision of the editors.

Reviewer 2 Report

All of my comments have been addressed, but please provide a clean version for the referee as there were so many changes in the manuscripts, its barely readible in this form.